# OpenReview forum: "Low-Rank Learning by Design: the Role of Network Architecture and Activation Linearity in Gradient Rank Collapse"
_NeurIPS.cc/2023/Conference — Submitted to NeurIPS 2023_

### Official Review · Reviewer_DvNk · 2023-07-06

**Soundness:** 3 good
**Presentation:** 3 good
**Contribution:** 3 good
**Rating:** 6
**Confidence:** 3

**Summary:**

The paper provides a comprehensive understanding of gradient rank in deep neural networks (DNNs) and how architectural choices and data structure affect gradient rank bounds. The authors highlight the emergence of low-rank learning as an inherent aspect of certain DNN architectures and propose a theoretical analysis to provide bounds for training fully-connected, recurrent, and convolutional neural networks. They also demonstrate, both theoretically and empirically, how design choices such as activation function linearity, bottleneck layer introduction, convolutional stride, and sequence truncation influence these bounds. The study not only contributes to the understanding of learning dynamics in DNNs but also provides practical guidance for deep learning engineers to make informed design decisions. The authors also discuss the phenomenon of "Neural Collapse," where linear classifiers within DNNs converge to specific geometrical structures during late-stage training, and highlight the role of geometric constraints in learning beyond this terminal phase.

**Strengths:**

This work theoretically analyzes the upper bound of the rank of gradients for different kinds of neural networks. The analysis can offer us great insight when we try to design new modules or activation functions. In addition, the authors provide a detailed practical analysis of the proposed methods.

**Weaknesses:**

For the experiments for validation of the bottleneck, it would be better to add experiments on convolutional neural networks to make the validation more sufficient.  For the analysis of the structure, it would be better to add analysis of some normalization techniques or regularization techniques.

**Questions:**

Please refer to Weakness.

**Limitations:**

Yes.

---

> ### Author Rebuttal · Authors · 2023-08-05
>
> Thank you for your positive feedback regarding our work on low-rank learning! We wholeheartedly agree that our approach to analyzing gradient rank can help in designing new modules and activation functions. We believe this positive direction of future work which you insightfully point out further encourages us regarding the importance of our submission to the NeurIps community and to the research community at large!
>
> ## Response to weaknesses
>
> ### Response to Request for Modern Architectures
> We completely agree that our analysis really shines when applied to modern, large-scale architectures! To this end, our original submission did include four large-scale analyses in our supplementary material, including two large-scale image-recognition models (ResNet, VGG) and two large-scale transformers for language modeling (BERT, XLM).  We would like to direct you to figures 3 and 4 in our supplementary material which demonstrate some of the effects of sequence length and image size in particular on the rank of gradients in modern, large-scale architectures.
>
> ### Response to Extension to Regularization Modules
> Thank you for your insightful comment that our analysis would indeed be augmented by an extension to normalization and regularization modules such as Batch Normalization and Layer Normalization. We think that because of the page limitations in this original submission, this would be a really fascinating endeavor for future work! In particular, an extension to mechanisms such as gradient clipping, L2/L1 regularization and other gradient-centric techniques would make for a fascinating body of work which could emerge from our initial work here!

---

### Official Review · Reviewer_TrYs · 2023-07-06

**Soundness:** 2 fair
**Presentation:** 1 poor
**Contribution:** 1 poor
**Rating:** 3
**Confidence:** 4

**Summary:**

The authors present an investigative study into the learning dynamics of neural networks, specifically low-rank neural networks. The motivation is that training dynamics of neural networks are not fully understood. Low-rank models have practical advantages (time / memory). This work provides a theoretical and empirical overview of the learning dynamics of neural networks with a focus on low-rank models.

**Strengths:**

The main strength of this paper is its significance, i.e. the attempt to explain learning dynamics of neural networks from both a theoretical and empirical perspective. This is a worthy and valuable endeavour which will be of great interest to the field if done correctly and thoroughly. Unfortunately, there are also significant weaknesses, as we will see, which weight heavily against the strengths.

**Weaknesses:**

There are a number of critical weaknesses present in this paper that severely reduce its value. Perhaps there is a valuable message in there but the presentation, clarity and writing make it impossible to recommend this as an impactful paper. Here are some of the weaknesses detailed:
- Flow and writing can be greatly improved. Very abrupt changes between sections (e.g. from 1 to 2). The sections do not have self-contained introductions to help readers orient themselves and understand the reasons and motivations for choices made.
- Related to the first point, formulae are presented without sufficient discussion. Many sections (notably 2 and 3) read like a series of statements rather than a well-constructed and clearly motivated argument. I encourage the authors to improve the flow to help the reader understand the context
- Formatting can be improved. What is *error*? Overview of results in lines 77-88 can be greatly improved with respect to formatting.
- Missing definition, what is BPTT in line 104?
- Lack of discussion on the connection between the presented theoretical and empirical results.
- Stated contributions are not found in the paper (effect of stride is in supplementary material and I could not find sequence truncation experiments anywhere).

**Questions:**

- Why are only leaky-ReLUs investigated, what is the motivation? And how does this differ from standard ReLUs?
- What is the theoretical connection to bottleneck layers bounding the entire network? Why is this limited to bottleneck layers and not the narrowest part of the network?
- What is the impact of training data on the learnt rank? Does the implicit dimension of the dataset influence the rank? With reference to [1]
- Why is the PDF formatted as an image and not as a standard NeurIPS PDF output? The links to section labels and reference links do not work.

[1] Li, C., Farkhoor, H., Liu, R., & Yosinski, J. (2018). Measuring the intrinsic dimension of objective landscapes. arXiv preprint arXiv:1804.08838.

**Limitations:**

The authors discuss only one limitation of their work: the relationship between tasks and low-rank emergence. However, there are a number of limitations that need to be discussed. Not only have the experiments scratched the surface of the possible signals from learning dynamics we can collect and paint us a limited picture of what is happening but the results themselves also make us pose new questions that remain undiscussed. Refer to my questions to the authors for some possible discussion points that would improve this paper greatly and help place it within the existing literature and highlight the most obvious next steps to build on this work for future studies.

---

> ### Author Rebuttal · Authors · 2023-08-05
>
> Thank you so much for recognizing the novelty and significance of our work! We appreciate your thorough critical feedback regarding the writing in particular. We have taken your feedback to heart and have worked diligently to ameliorate these issues while still working within page limitations. Particular technical issues such as links in the PDF not working, which surprised us as well, have also been fixed.
>
> We also thank you for bringing up some issues regarding stated contributions which you could not find in the main body of work. As mentioned in the main text, much additional work was included in supplementary material due to page limitations. In our specific responses below we have directed you to the particular figures in our original supplementary material.
>
> ## Response to Weaknesses
>
> ### Response to Issues of Writing Quality
>
> Thank you for your feedback regarding the overall flow of the work! We wholeheartedly agree that section introductions and motivating context can really help ground the reader. We were forced to cut some of the original language to this effect in order to make the page limit; however, we agree that the clarity of our paper has suffered because of these cuts. We have worked to provide some additional context particularly in the methods sections where we believe this is most helpful in grounding our presentation. We will continue to revise the text for flow and clarity where possible!
>
> ### Response to Claim of Results not Included
>
> As far as we have found after review, we did not make any claims that were not included in the main text or the supplementary material. The experiments we performed with differing sequence length are included for the BERT architecture on Wikitext in Figure 4 of the supplementary material, which was indeed included with the original submission. We understand that the language surrounding stride in particular may have been confusing, as our experiment with Image Size in CNNs (also included in the supplement) covers the same underlying mechanism  - i.e. the length over the sequence over which parameter tying occurs. We have changed the language in our introduction to make this more apparent.
>
> ## Response to Questions
>
>   1. Great question! Leaky-ReLUs are equal to ReLUs when the slope of the nonlinearity is 0, so when we increase the level of linearity, we are slowly increasing from a fully linear activation to a fully-nonlinear, standard ReLU. The reason we investigate Leaky-ReLUs in particular is that it gives us a clear theoretical bridge between the fully-linear network space (which has been investigated in many previous theoretical works from Saxe et al. for example) to the more-widely-applied DNNs with nonlinearities.  Additionally, because Leaky-ReLUs (and RelUs) are piecewise linear, we are able to provide analytical bounds on their effect on the rank!  We do believe that there is room for future work on other nonlinear activation functions; however, the standard machinery of Linear Algebra does not apply to these nonlinear functions. There is some work in functional analysis and nonlinear control theory which we believe may contribute some analogous notions to Singular Values (and thus to rank) for general nonlinear functions. We have included some figures empirically investigating rank effects of other nonlinearities in our rebuttal, and hope this excites you for the future directions of this work!
>   2. This is a great question!  We call the narrowest part of the network a bottleneck. For example in an MLP, the layer with the smallest number of neurons will be one. Inequality 2 in the submission demonstrates how such a bottleneck affects the rank of the entire network. Bottleneck layers at the input and output will also serve as bottlenecks on the entire rank of the network. For example in MNIST classification with a linear network, the rank of the gradients will be limited to 10 throughout unless there is a layer with number of neurons smaller than 10.
>   3. This is a great question, which we address in a figure (Figure 1)  in the supplement! Indeed, the rank of the input will affect gradient ranks - we show with a low-rank embedded Gaussian in the supplement that this will pop up in gradients of linear networks. We have checked the provided reference and although it uses a very different mechanism, we have cited this work when we mention corresponding results.
>   4. Thank you for pointing out this technical glitch in our submission! We believe this occurred due to a resave of the document outside of the latex software, and we have fixed this in our revised document
>
> ## Response to Limitations
> While we agree that there are many more implications of our work to explore, we do discuss a number of limitations to our work beyond what the reviewer has pointed out. Our analysis focuses only on Leaky-ReLU and ReLU activations, and further work here is discussed several points throughout the paper.  Additionally, we acknowledge that there is ample room for extension to regularization techniques such as Batch Normalization, Layer Normalization and more, which we treat as evidence of potential impact of our work. We have also discussed a possible connection to Neural Collapse which is discussed in a limited way throughout the paper. We realize that more language in the discussion section could be included to reemphasize these points which appear throughout our work, and we have extended our discussion as much as possible while staying within the page limit. Additionally, we would like to direct the reviewer to the supplementary material that includes some additional discussion linking our work to previous literature on ReLU singular values and Neural Collapse. Per your insightful recommendation regarding the detection of rank in input data, we have provided a citation to the work as recommended, and we believe there is potential for further work connecting our work to the study of rank in objective landscapes.

---

> > ### Comment · Reviewer_TrYs · 2023-08-20
> >
> > Thank you for the comprehensive response to my review.
> >
> > I find the author’s proposed changes to be positive for the overall paper. However, I don’t find all my concerns have been alleviated, particularly with respect to many of the critical aspects of the paper which are still slightly “out of view” in the supplementary.
> >
> > Having said that I feel the proposed changes are significant enough to warrant an improved rating, which I will do by editing my original review rating.

---

### Official Review · Reviewer_ciGq · 2023-07-06

**Soundness:** 3 good
**Presentation:** 3 good
**Contribution:** 3 good
**Rating:** 5
**Confidence:** 2

**Summary:**

This paper studies the gradient rank of DNNs and examine how certain design choice, in particular architectural choices of the model and the structure of the data affect the gradient rank bounds. The paper mainly focuses on theoretical results with some empirical experiments to validate their empirical claims. The paper begins by studying the gradient rank of deep linear models before proceeding to explore leak-ReLU networks. They also extended their analysis to consider convolution and recurrent layers.

**Strengths:**

The paper seems fairly novel, as I have not seen anyone explore the gradient rank although I think it is important as it can tell us more about the learning dynamics of neural networks. The theoretical analysis is simple, and this is advantageous in my opinion. One of the strengths of the paper is they have analysed a few neural network variants such as recurrent, convolutional and leaky relu networks. This demonstrates how general their analysis can be. The experiments are good and test well the bounds.

**Weaknesses:**

it is not clear how to generalise the analysis to more complex non-linearities and can thus at most only seem to study them near a limited range of input values (such as near zero) where the network behaves more linearly. The analysis also doesn't look at more modern architectures like transformers.

**Questions:**

DNNs are trained using gradient descent. It is possible to write the parameters of the network as a weighted sum of gradients, since we can bound the rank of the gradients with your analysis can we use that to bound the rank of some of the layers in a DNN.

**Limitations:**

Authors have adequately addressed and there is no clear potential negative societal impacts.

---

> ### Author Rebuttal · Authors · 2023-08-05
>
> We would like to thank you for your positive review of our work! We wholeheartedly agree that one of the strengths of our analysis is the simplicity at its core, which we believe can help lead to better intuition in the growing study of learning dynamics in deep neural networks!
>
> ## Response to Generalization to other Nonlinear Functions
>
> Your feedback regarding other nonlinear functions is quite insightful! As you point out,our analysis would not immediately generalize to any nonlinearity that is not piecewise linear. Our primary reason for not taking on general nonlinear activations is that the machinery of linear algebra (particularly the singular value decomposition) does not apply to non-linear functions, and so more mathematical groundwork is needed to motivate any discussion of general nonlinearities. Our work newly recognizes that any piecewise nonlinear function can be analyzed using the standard linear-algebraic machinery; however, to move forward to more general nonlinear functions, new mathematical machinery is needed perhaps drawing on previous work in functional analysis of nonlinear functions, or in control theory of nonlinear operators.
>
> A more precise way of putting this limitation is that our analysis can deal with any nonlinear function that is piecewise linear (even “very nonlinear” functions such as standard ReLUs). What our results show though is that even a small amount of nonlinearity in a leaky-ReLU will result in ncreasing the rank up to an upper bound. We believe that there is an exciting frontier which takes on general nonlinear functions; however, because the groundwork needed more significantly departs from previous work analyzing rank, we believe this is better suited for future work.
>
> We would like to refer you to the figures included as part of our rebuttal for an initial empirical investigation of rank-based phenomena in other nonlinear activation functions! We believe there is much more to investigate here, and our initial empirical findings demonstrate some fascinating behavior that begs for thorough theoretical review!
>
> ## Response to Request for Modern Architectures
>
> Regarding your point that our analysis doesn’t look at more modern architectures - we would like to point you to our supplementary material where we do perform empirical studies of large-scale networks including two large-scale CNN-based models for image recognition, and a large transformer model for language modeling.
>
> Our theoretical analysis generalizes quite readily to most of the parts of these models as we do discuss the role of parameter tying in CNNs and RNNs (which includes Transformers). We do leave the analysis of auxiliary modules such as Batch and Layer normalization for future work; however, our empirical results on models which do include batch normalization indicate that our analysis still applies.

---

### Official Review · Reviewer_nQBm · 2023-07-07

**Soundness:** 2 fair
**Presentation:** 2 fair
**Contribution:** 2 fair
**Rating:** 3
**Confidence:** 3

**Summary:**

This paper explores the collapse of gradient rank in DNNs during training. Based on the simple linear network, the authors theoretically examinate the rough upper bound of the gradient rank for simple MLP, current network and CNN. Furthermore, they analyze the numerical effect of rank on Leaky-ReLU activation.

To verify their results of the rank bounds, experiments are conducted with on two synthetic datasets and several real datasets (CIFAR10 and Tiny-ImageNet for computer vision and WiKiText for NLP) across pure linear networks and smaller-sized version of popular architectures, e.g., ResNet16, VGG, BERT, etc.

Experimental results show that 1. Linear bottleneck layer architecture reduces the gradient rank; 2. the gradient rank of recurrent networks and CNNs is proportion to the size of sequence; and the negative slope of leaky-ReLU activation is related to the amount of gradient rank restored.

**Strengths:**

The paper demonstrates a high level of originality in its exploration of gradient rank in linear network, recurrent networks and CNN and extra analysis of Leaky-ReLU effect.

**Weaknesses:**

1. Poor writing quality: This paper has not been properly polished before submission, and there are numerous writing issues that have not been carefully checked. For instance, in line 132, the term "phi" should be "\phi". Additionally, in line 217, the notation "FIGURE1left" should be revised for clarity. In line 218, there are two issues: the terms "FIGURE1right" and "In figure" should be corrected ("in figure"?). Moreover, a missing "." in line 235 needs to be addressed. Another concern is the inconsistent section references throughout the paper, such as "sec 3.2.2" in line 226 and "&3.3.1" in line 236. Overall, this article resembles a draft rather than an official submission, and it requires further polishing and thorough review.

2. Unfollowed style: the bibliography style seems not to follow the provided template and please check your article format, because I cannot search and select your words in the main context and cannot jump to the referred figures.

3. Lack of practical network experiment: although the theoretical analysis is mainly conducted on the linear network and experiments are provided for 3-layer linear MLP and RNN network, more experiments and further analysis on popular networks are needed for providing insight for the understanding the gradient rank in practice.

4 Lack of citation and explanation for BPTT.

**Questions:**

Because the paper is not well-written, most question is for clarification.

1. what does the "disjoint variable" mean in line 171?

2. what does the "original bound" mean in line 244?

3. what does the "restore" refer to in line 246-255?

**Limitations:**

No potential negative societal impact of their work

---

> ### Author Rebuttal · Authors · 2023-08-05
>
> We would like to sincerely thank you for your review of our work! We are especially encouraged by your recognition of the originality of our work—we believe that this rank-based approach to investigating learning theory in Deep Neural Networks is extremely promising, and the results we share in this work are an important first step in setting groundwork for further investigations on the role of rank in gradient dynamics. It is our belief that the weaknesses you perceived in our work are in part addressed by referring to results which we did include in the supplementary materials of the original submission, and the writing issues you point out have been easily addressed by thorough editorial review and polish!
>
> ## Response to weaknesses
>
> ### Response to Issues on Writing Quality
>
> We would like to thank you for your feedback regarding issues with the writing in our original submission! We agree that clear and polished writing is a vital part of communication in science, and we have taken your feedback as a clear indication that this part of our submission needs significant improvement.  The particular issues you have correctly pointed out have been fixed in the revised document, and we are currently continuing to review the submission for any further issues with the writing.  Additionally, we have fixed the technical issue with the PDF which was preventing links in the bibliography to work.
>
> ### Response to Issues with Bibliography Style
>
> Thank you so much for pointing out the technical issue with our submission! Although we did use the correct template*, we believe a resave of the document during the revision process somehow broke links to the bibliography and figures. This is fixed in the revision which we have uploaded here, and we thank you again for bringing it to our attention!
>
> Regarding the citation style - we did use the provided NeurIps template; however, we would like to refer to the style guide regarding our choice of citation style: "Any choice of citation style is acceptable as long as you are consistent. It is permissible to reduce the font size to \verb+small+ (9 point) when listing the references." From the style guide https://media.neurips.cc/Conferences/NeurIPS2023/Styles/neurips_2023.tex
>
> ### Response to Perceived Lack of Large-Scale Experiments
>
> Thank you for your feedback and concern that you did not see any large-scale experiments in the main text of the paper. We are of the same mindset that such large-scale experiments are important, and indeed we included several large-scale analyses on modern architectures as part of the original submission! In the second-to-last paragraph of our Empirical Methods section (section 4 — lines 198-207) we outline the experiments which were performed, which include two popular image-recognition data sets (Cifar10/Tiny-ImageNet) and a large-scale language modeling data set (WikiText). We implemented two popular image-recognition CNN (ResNet16/VGG11) and language-modeling transformer (BERT, XLM) architectures. Because these architectures are quite large, and the results simply reinforce the proof-of-concept results in the main text, we included the results as part of our supplementary materials. Because NeurIPS requires separate documents for the main text and supplementary materials, we cannot link to the figures directly; however, we have made the references to these experiments more clear. We would encourage the reviewer to look through the supplementary material, as we believe these results do indeed help to reinforce the importance of our work!
>
> ### Response to Citation for BPTT
>
> Thank you for catching our omission of the full phrase “Back-Propagation through Time”! We have added the full phrase as should have been originally included! While there are several possible candidates for citing such a popular optimization paradigm, we have elected to provide a citation Mozer et al. 1995 which is one of the foundational works on BPTT.
>
> ## Response to Questions
>
>   1. We believe this reviewer is referring to the “Adjoint Variable” on line 171, rather than “disjoint” which we cannot find anywhere in the text. The adjoint variable is the name given to the statistics during the backward phase of Reverse-Mode Auto-Differentiation, which in this case is the partial derivative on layer output. This is defined in the “Theoretical Methods” section after line 99, and is a commonly-used term in the Auto-Differentiation literature.
>   2. The “original bound” here refers to the boundary under which singular values numerically computed from the output of a leaky-ReLU activation will no longer contribute to the rank. We agree that this language is confusing, and we have changed the text to refer to this as the “rank computed empirically from the output of the activation”. This better demonstrates that our theoretical result is able to exactly predict this bound which previously could only be computed using numerical methods.
>   3. In this paragraph, when we say a given nonlinearity is or is not “restoring the rank” we mean to say that the resulting output is no longer as rank-deficient as the original matrix. Therefore, when we say that ReLU activations “do not fully restore gradient rank” we simply mean that the resulting output matrix is still not full-rank. We hope this has clarified what was meant here, and we have changed the wording to “restoring  to full rank” to hopefully make this more clear to future readers.

---

### Author Rebuttal · Authors · 2023-08-05

We would like to thank the reviewers for taking the time to critically evaluate our submission on Low-Rank Learning in Deep Neural Networks! All four of the reviewers recognized the novelty and importance of the work, and we are encouraged by this positive feedback! Our analysis provides important groundwork for analyzing the role of gradient rank in deep learning, and we believe the submission would be important to the NeurIps Community at large.

Much of the critical feedback in this process has focused on technical issues with the writing in the original document, including some broken links, typos, obfuscated language, and need for thorough polishing. We have fixed all specific issues pointed to by reviewers, and are continuing to thoroughly review our manuscript to better increase clarity and improve overall flow. The paper has already been improved significantly by reviewer feedback, and we believe any remaining issues can be easily fixed for the camera-ready version.

All four of our reviewers brought up concerns that our submission lacked large-scale experiments; however, as we mention in the specific responses, four large-scale, modern architectures were included as part of the original submission, with several paragraphs in our methods and results referring to their inclusion as such. Space limitations demand that these figures remain as part of the supplementary material; however, we are thankful for this feedback as it has allowed us to better highlight the supplementary material in the main text with clearer language and more direct reference.  In general, we believe that **all critique regarding content is addressed by referring to the supplementary materials** included as part of the original submission.

One fascinating question brought up by several reviewers asks about the extension of this work to general nonlinear functions. Our analysis breaks new ground in deep learning theory in its ability to apply to any nonlinear activation which is piecewise linear; however, we wholeheartedly agree that an extension to general nonlinear functions would be a significantly useful extension for future work. The primary reason for omitting that from this submission is that the standard machinery of linear algebra does not apply to general nonlinear functions, and thus significantly more mathematical groundwork is needed to investigate the connection to rank.

Despite our reasons for not pursuing other nonlinear activations  as part of this submission, we do believe however, that our submission here is important first step in this future direction. As part of this rebuttal we have included a figure showing how different nonlinearities affect the magnitude of the singular values of a given low-rank linear transformation. In this figure, each row represents a different activation function, and each column $k$ represents the magnitude of the $k$th singular value as a function of the magnitude of the top 2 singular values. For this demonstration, we use a transformation with an underlying rank of 2, and we see that different nonlinear activations affect the spectrum quite uniquely as functions of the magnitude of the original top 2 eigenvalues. We hope this demonstrates how fascinating but subtle this rank-based analysis of nonlinear activation can be, and reinforces the importance of our work as an initial step in a fruitful direction for future research.

Overall we are encouraged by the feedback from this team of reviewers! All of these reviewers acknowledge the originality and importance of the work, any issues with content are mitigated by reference to supplementary material. We have work to make references to the supplement more clear in the main body of text to aid future readers. We have fixed all specific issues with writing brought up by the team of reviewers, and we are continuing to polish the composition so that the communication of our work is clear for the NeurIps and research  communities at large.

Thank you to the reviewers for your thorough feedback! Your critique and questions have significantly improved the quality of our submission, and we are now even more confident in the value this work would have to the research community!

---

### Decision · Program_Chairs · 2023-09-21

**Decision:**

Reject

**Comment:**

This paper explores an interesting phenomenon on the collapse of gradient rank in DNNs during training.
However, the paper in the current form has several limitations:
1. The presentations of the work need significant improvement
2. More experiments and discussion on the connection between theory and practice.

We encourage the authors to incorporate the feedbacks from the reviewers for future submission.